# Identification of the Inner Cell Mass and the Trophectoderm Responses after an In Vitro Exposure to Glucose and Insulin during the Preimplantation Period in the Rabbit Embryo

**DOI:** 10.3390/cells11233766

**Published:** 2022-11-25

**Authors:** Romina Via y Rada, Nathalie Daniel, Catherine Archilla, Anne Frambourg, Luc Jouneau, Yan Jaszczyszyn, Gilles Charpigny, Véronique Duranthon, Sophie Calderari

**Affiliations:** 1BREED INRAE, UVSQ, Université Paris-Saclay, 78350 Jouy-en-Josas, France; 2Ecole Nationale Vétérinaire d’Alfort, BREED, 94700 Maisons-Alfort, France; 3Institute for Integrative Biology of the Cell (I2BC), UMR 9198 CNRS, CEA, Paris-Sud University F, 91190 Gif-sur-Yvette, France

**Keywords:** preimplantation embryo, diabetes, DOHaD, rabbit

## Abstract

The prevalence of metabolic diseases is increasing, leading to more women entering pregnancy with alterations in the glucose-insulin axis. The aim of this work was to investigate the effect of a hyperglycemic and/or hyperinsulinemic environment on the development of the preimplantation embryo. In rabbit embryos developed in vitro in the presence of high insulin (HI), high glucose (HG), or both (HGI), we determined the transcriptomes of the inner cell mass (ICM) and the trophectoderm (TE). HI induced 10 differentially expressed genes (DEG) in ICM and 1 in TE. HG ICM exhibited 41 DEGs involved in oxidative phosphorylation (OXPHOS) and cell number regulation. In HG ICM, proliferation was decreased (*p* < 0.01) and apoptosis increased (*p* < 0.001). HG TE displayed 132 DEG linked to mTOR signaling and regulation of cell number. In HG TE, proliferation was increased (*p* < 0.001) and apoptosis decreased (*p* < 0.001). HGI ICM presented 39 DEG involved in OXPHOS and no differences in proliferation and apoptosis. HGI TE showed 16 DEG linked to OXPHOS and cell number regulation and exhibited increased proliferation (*p* < 0.001). Exposure to HG and HGI during preimplantation development results in common and specific ICM and TE responses that could compromise the development of the future individual and placenta.

## 1. Introduction

The worldwide prevalence of metabolic diseases such as diabetes is increasing at an alarming rate [1]. In 2021, the International Diabetes Federation estimated that 1 in 10 adults live with diabetes [1]. Type 2 diabetes (T2D) is a chronic metabolic disease characterized by hyperglycemia, insulin resistance, and/or impaired insulin secretion and accounts for 90% of diabetes cases [1]. In prediabetes and the early stages of T2D, impaired glucose tolerance, or hyperglycemia, is accompanied by compensatory hyperinsulinemia due to decreasing insulin sensitivity [1,2]. Unfortunately, these first signs of metabolic dysregulation are often asymptomatic, resulting in nearly half of T2D patients going undiagnosed and untreated [1]. Known before as adult-onset diabetes, the prevalence of T2D is increasing in younger people, including women of childbearing age [1,3]. Type 1 diabetes (T1D), an immune-related disease characterized by the destruction of insulin-producing cells, affects a young population [4]. In T1D, the glucose-insulin axis is disrupted. Insulin is no longer produced, and insulin-stimulated glucose uptake is reduced, resulting in persistent hyperglycemia [4]. One in six pregnancies is estimated to be affected by hyperglycemia [1]. Exposure in utero to a perturbed glucose-insulin homeostasis increases the risk of birth defects and metabolic deregulations such as enhanced growth, higher fasting glucose, and lower insulin sensitivity in the offspring [5]. These metabolic dysregulations can be maintained throughout the life course of the individual, making it prone to developing cardiometabolic diseases such as obesity and T2D [3]. This is described by the Developmental Origins of Health and Disease (DOHaD) concept, which highlights that exposure to a suboptimal environment during critical periods of development predisposes the offspring to poor health later in life [6]. One key period of development sensitive to environmental insults is the preimplantation stage [7]. During the preimplantation stage, embryos undergo tightly regulated essential events such as the maternal-to-zygotic transition with the transcriptional activation of the embryonic genome (EGA) and the first lineage specification giving rise to the inner cell mass (ICM), the progenitor of the embryo proper, and the trophectoderm (TE), the progenitor of the embryonic portion of the placenta [8]. To sustain their development, embryos take advantage of the nutrients and growth factors present in the oviduct and uterine fluid [9,10]. The composition of these fluids varies according to maternal metabolic and hormonal status, as is the case for glucose and insulin, whose concentrations depend on maternal circulating plasma concentrations [9,10,11]. Preimplantation embryos are sensitive to perturbations in their surrounding microenvironment [7,8,11]. Variations in the environment of the early embryo, even restricted to the preimplantation period, result in irreversible defects in the adult offspring [12]. Studies in vivo and in vitro have demonstrated the susceptibility of preimplantation embryos to changes in glucose or insulin levels [13]. In diabetes-induced rabbit and mouse models, preimplantation embryos exposed to hyperglycemia resulted in perturbed insulin-mediated glucose metabolism, decreased glucose transport and utilization, reduced developmental competence and cell numbers, and increased apoptosis in the ICM [14,15,16]. In in vivo animal models, severe hyperglycemia was obtained by the chemical destruction of pancreatic β-cells, thus mimicking type 1 diabetes. Nevertheless, because insulin secretion was reduced or absent in these animals, frequent insulin injections were needed, which may have resulted in oscillating insulin levels in the intrauterine environment [15]. Unfortunately, insulin levels were not quantified in these studies; thus, it is impossible to identify whether the phenotypes described were the result of hyperglycemia or the combination of hyperglycemia and insulin. In vitro, exposure to high glucose alone led to impaired blastocyst development, reduced total cell numbers, decreased glycolytic activity, decreased insulin sensitivity, perturbed TE differentiation, and impaired capacity of trophoblast outgrowth in vitro—a marker of implantation potential [11,14,17]. Preimplantation embryos are exposed to insulin, which is present in the oviductal and uterine fluids at concentrations that depend on maternal insulin levels [14]. The extent of the cellular and molecular responses to insulin in early embryos has been less investigated [13]. Glucose and insulin, through the activation of signaling and metabolic pathways, are closely related [18]. In preimplantation embryos, glucose is used as an energy source, reaching the highest consumption rate at the blastocyst stage [19,20]. Furthermore, insulin receptors and insulin-responsive glucose transporters are expressed in mouse, rabbit, and human preimplantation embryos [21]. 

We hypothesized that the deregulation of glucose and insulin homeostasis present in an increasing number of women impacts the preimplantation embryo. Functionally different from the blastocyst stage, ICM and TE differ in their epigenetic, transcriptomic, and metabolic programs [20,22,23]. We hypothesized that exposure to this glucose-insulin altered environment affects ICM and TE differently and induces short- and long-term consequences not only in the future individual but also in the future placenta, a central element for fetal nutrition regulation, and whose structure and/or function adapt to suboptimal in utero environments [15,24,25]. Hence, to investigate the effects of high glucose and/or high insulin on preimplantation development, we used the rabbit model, a model with preimplantation development (i.e., EGA timing, gastrulation morphology), glucose metabolism at early stages, and a placental structure close to that of humans [26]. We established a model of one-cell stage rabbit embryos developed in vitro until the blastocyst stage with supplementation of glucose, insulin, or both to recreate a moderately hyperglycemic and/or hyperinsulinemic environment [11,14,17,27] and addressed the specific gene expression responses of the ICM and TE. 

## 2. Materials and Methods

### 2.1. Embryo In-Vitro Development 

New Zealand White female rabbits (INRA line 1077) were superovulated as previously described [28] and mated with New Zealand White male rabbits. At 19 h post-coïtum (hpc), does were euthanized, and one-cell embryos were recovered from oviducts by flushing with phosphate buffer saline (PBS, Gibco, Thermo Fisher Scientific, Waltham, MA, USA). One-cell embryos were sorted in M199 HEPES (Sigma-Aldrich, Saint-Louis, MO, USA) supplemented with 10% fetal bovine serum (FBS, Gibco) and rinsed in Global medium (LifeGlobal Group, Guilford, CT, USA) supplemented with 10% human serum albumin (HSA, LifeGlobal Group). Embryos were then placed in 10 µL microdrops of Global-10% HSA medium supplemented with either glucose (Sigma-Aldrich G6152) and/or insulin (Sigma-Aldrich I9278) and covered with mineral oil (Sigma-Aldrich M8410) for a 72h culture at 38 °C, 5% CO_2_, and 5% O_2_ until the blastocyst stage. Four experimental groups were designed: Control (CNTRL): 0.18 mM of glucose without insulin; high insulin (HI): 0.18 mM of glucose and 1.7 μM of insulin; high glucose (HG): 15 mM of glucose without insulin; and high glucose and high insulin (HGI): 15 mM of glucose and 1.7 μM of insulin. After 72 h of culture, to determine the embryo’s developmental competence in each group, embryos were classified into three categories: (i) arrested embryos; (ii) compacted embryos; (iii) blastocysts or cavitated embryos. The rate of arrested embryos (developmental arrest), compacted embryos, and blastocysts/cavitated embryos reported in percentage was calculated in fifteen to twenty-nine independent experiments from the total of one-cell embryos placed in culture. Blastocysts were recovered to proceed to ICM and TE isolation by moderate immunosurgery. To remove the zona pellucida, blastocysts were incubated for 1–3 min in 5 mg/mL Pronase (P5147, Sigma-Aldrich). Embryos were next incubated in anti-rabbit goat serum (R5131 Sigma-Aldrich) for 90 min at 37 °C and then incubated in guinea pig complement (S1639 Sigma-Aldrich) for 20 sec. The ICM was mechanically isolated from the TE by pipetting with a small-bore glass pipette (60–70-µm diameter). To clean the ICM to limit any contamination, several back and forth injections into the glass pipette were perfomed. ICM and their corresponding TE were then immediately stored at −80 °C for RNA sequencing analysis or fixed for microscopic analyses.

### 2.2. RNA Sequencing 

ICM and their corresponding TE originating from the same blastocysts were used. Only one biological replicate from the HI group did not include the corresponding TE due to low total RNA quality. Total RNA was extracted from three biological replicates per culture condition, corresponding to pooled samples (*n* = 11–16 ICM or TE per replicate) using the Arcturus PicoPure RNA Isolation Kit (Applied Biosystems Life Technologies, Waltham, MA, USA). RNA quality was assessed using RNA 6000 Pico chips with an Agilent 2100 Bioanalyzer (Agilent Technologies, Santa Clara, CA, USA). All extracted samples had an RNA Integrity Number (RIN) ≥ 8 value. Seven hundred and fifty pictograms of total RNA were used for amplification using the SMART-Seq V4 ultra-low input RNA kit (Clontech, Takara, Saint-Germain-en-Laye, France) according to the manufacturer’s recommendations with nine PCR cycles for cDNA pre-amplification. The cDNA quality was assessed with the Agilent Bioanalyzer 2100. Libraries were prepared as previously described [29]. Reads were mapped to the rabbit transcriptome reference (Ensembl 98 Oryctolagus cuniculus 2.0) using the splice junction mapper TopHat (v2.1.1) associated with the short-read aligner Bowtie2 (v2.3.4.1). To generate the gene count table, featureCounts (v1.6.0) was used. Hierarchical clustering was computed as previously described [29]. Data normalization and single-gene level analysis of the differential expression were performed using the DESeq2 package (v1.28.1) [30]. Differences were considered significant for adjusted *p*=values (Benjamini-Hochberg) < 0.05 and when the normalized expression counts were more than 20 in two of the three biological replicates. Heatmaps were generated with the pheatmaps R package (v1.0.12), with the z-score calculation of the normalized expression counts obtained with DESeq2. Logarithm 2 Fold Change (Log2FC) of differentially expressed genes (DEG) was used to generate horizontal bar plots with R studio software (v1.2.5019). InteractiVenn [31] software was used for Venn diagram generation. Functional annotation of DEG with their associated Gene ontology (GO) Biological Process (BP) terms was performed using DAVID [32] (v6.8). Gene Set Enrichment Analysis (GSEA) [33] was performed using the GSEA Java Desktop application (v4.0.3) from the Broad Institute. Enrichment analysis was calculated using the normalized expression counts obtained with DESesq2 and the Molecular Signature Database (MSigDB, v7.0) gene set collections (Hallmarks [34], KEGG (Kyoto Encyclopedia of Genes and Genomes), Reactome [35], and GO BP [36,37]) by gene-set permutation. Gene sets were considered significant when the false discovery rate (FDR) was less than 0.05. Enrichment analysis results were analyzed with the R package SUMER [38] (v1.1.5) for the reduction of redundancy and condensation of gene sets. For cluster visualization, the clusterMaker2 [39] plugin from Cytoscape [40] (v3.8.2) was used.

### 2.3. Quantification of Total Cell Number in Whole Embryos

To quantify the total cell number in whole embryos, DAPI staining was assessed in in vitro-developed blastocysts. Blastocysts were recovered from in vitro culture, and the zona pellucida was removed as detailed above. Blastocysts were fixed in 4% paraformaldehyde (PFA, EMS) in PBS at room temperature (RT) for 20 min. Permeabilization was performed with 0.5% Triton X-100 (Sigma-Aldrich) in PBS with 0.5% polyvinylpyrrolidone (PVP) for 1 h at 37 °C in a humidified chamber. DNA was counterstained with 0.2 mg/mL DAPI (Invitrogen) in PBS for 15 min at RT. Blastocysts were analyzed by an inverted ZEISS AxioObserver Z1 microscope (Zeiss, Rueil Malmaison, France) equipped with an ApoTome slider (Axiovision software 4.8) using a 20X objective and a z-distance of 1.5 μm between optical sections at the MIMA2 platform (https://doi.org/10.15454/1.5572348210007727E12, accessed on 4 October 2022). The total number of DAPI-labeled nuclei was quantified manually using ImageJ software (1.53.j). Each condition was analyzed in five to nine independent experiments.

### 2.4. Quantification of Apoptotic and Proliferating Cells in ICM and TE 

To quantify apoptotic and proliferating cells in ICM and TE, we first considered distinguishing the ICM and TE on whole embryos by immunostaining of known lineage-specific markers. CDX2 (CDX-2-88, Biogenex, Fremont, CA, USA), SOX2 (ab97959, Abcam, Cambridge, UK), and NANOG (14-5761-80, Invitrogen, Thermo Fisher Scientific, Waltham, MA, USA) antibodies were tested, but none showed sufficient specificity to consider differential counting (data not shown). Thus, determination of apoptotic and proliferating cell numbers was performed on isolated ICM and TE. 

Detection of apoptotic cells was performed using the DeadEnd Fluorometric TUNEL System (Promega, Madison, WI, USA) in two to six independent experiments. Isolated ICM or TE were fixed in 4% PFA in PBS at RT for 20 min. Permeabilization was performed with 0.5% Triton X-100 in PBS with 0.5% PVP for 1 h at 37 °C in a humidified chamber. After rinsing in PBS with 0.5% PVP, a second fixation was performed in 4% PFA and 0.2% glutaraldehyde for 15 min at RT. As a positive control, ICM and TE were treated with 2 units of RQ1 RNase-free DNase (Promega) for 30 min. The TUNEL reaction was performed according to the manufacturer’s directions. DNA was counterstained with 0.2 mg/mL DAPI in PBS for 15 min at RT.

Detection of proliferating cells was performed using the Click-iT^®^ Edu Imaging Kit (Fisher Scientific, Waltham, MA, USA) in at least three independent experiments. Briefly, the zona pellucida was removed as detailed above, and then blastocysts were incubated with 10 µM EdU for 15 min at 38 °C, 5% CO_2_, and 5% O_2_. ICM and TE separation were performed by moderate immunosurgery. ICM and TE were fixed with 4% PFA at RT for 20 min. EdU detection was performed according to the instructions provided by the manufacturer. DNA was counterstained with 0.2 mg/mL DAPI in PBS for 15 min at RT.

ICM and TE were analyzed by an inverted ZEISS AxioObserver Z1 microscope equipped with an ApoTome slider using a 20X objective and a z-distance of 1.5 μm between optical sections at the MIMA2 platform. The number of DAPI-labeled nuclei, TUNEL-positive nuclei, and EdU-positive nuclei were quantified manually using ImageJ software. 

### 2.5. Statistical Analysis 

Statistical analysis was carried out using the generalized linear mixed-effects model (GLMM) with the glmer function and the lme4 R package (v1.1-28). The total cell number was analyzed using the linear mixed-effects model (LMM) using the lmer function. The glucose and insulin concentrations were considered as fixed effects. No significant interaction between glucose and insulin was detected. The models applied in the analysis of developmental competence and total cell number did not include the interaction of glucose and insulin. The in vitro culture experiments and rabbits were considered to have random effects. Estimated marginal means (emmeans, also known as least-squares means) and post-hoc tests between conditions were performed using the emmeans R package (v1.7.3) with the emmeans and pairs functions. Results are shown as emmeans with standard errors. Differences were considered significant when *p*-values were < 0.05.

## 3. Results 

To determine the effect of high glucose and/or high insulin during preimplantation development, one-cell rabbit embryos were cultured in vitro under control (CNTRL), high insulin (HI), high glucose (HG), or high glucose and high insulin (HGI) until the blastocyst stage (Figure 1). To evaluate the effect of these conditions on developmental competence, the mean percentage of arrested embryos, compacted morula, or expanded blastocysts at the end of the 72 h culture period was determined (Table 1). On blastocysts, ICM and their corresponding TE were separated by moderate immunosurgery (Figure 1), and specific transcriptomic responses to high glucose and/or high insulin were explored by RNA-sequencing. RNA-seq of three biological replicates per culture condition generated 102–145 million raw reads per sample. Clustering of the transcriptome datasets by Euclidean distance revealed a clear separation between the ICM and TE regardless of the condition (Figure 2A). Without excluding minimal contamination, these results underline the successful separation of these two compartments by immunosurgery. Principal component analysis (PCA) was performed separately on ICM (Figure 2B) and TE (Figure 2C) transcriptomic data. Comparison to the CNTRL resulted in the identification of differentially expressed genes (DEG) between ICM or TE from embryos developed in HI, HG, or HGI (Figure 3 and Appendix A). Functions of the identified DEGs were explored using GO terms annotations (Appendix A). To determine coordinated gene expression changes, we analyzed the gene expression datasets using GSEA with the Hallmarks gene set collections, KEGG, Reactome, and GO BP databases (Appendix A). Enrichment analysis results were then analyzed with SUMER for gene set condensation. The following paragraphs will describe the identified effects of high insulin or high glucose alone and then in combination in the ICM and TE of exposed embryos. 

### 3.1. Impact of High Insulin In Vitro Exposure 

The developmental competence of HI embryos showed no significant differences when compared to the CNTRL condition (Table 1). Quantification of total cell number by DAPI staining did not show significant changes in HI (262 ± 12, *n* = 54) versus CNTRL (240 ± 7, *n* = 76) blastocysts (*p* < 0.05, Appendix A).

#### 3.1.1. In ICM, High Insulin Induced Changes in Cellular Energy Metabolic Pathways

Transcriptome analysis by PCA and hierarchical clustering did not show a clear separation between HI ICM and CNTRL ICM (Figure 2). Differential analysis of HI ICM versus CNTRL ICM transcriptomes identified 10 DEG (3 overexpressed and 7 underexpressed) (Figure 3 and Appendix A). GSEA identified 37 significant positively enriched pathways (2 Hallmarks, 3 KEGG pathways, 14 GO BP, and 18 Reactome gene sets) and 8 negatively enriched (5 Hallmarks and 3 GO BP) pathways (Figure 4A and Appendix A).

Enriched pathways included gene sets implicated in translation and oxidative phosphorylation (OXPHOS) (Figure 4A). Analysis of DEG and enrichment results in HI ICM transcriptomes compared to CNTRL ICM pointed out the perturbation of transcription and translation. Gene-by-gene statistical analysis identified DEG implicated in the regulation of transcription as *RC3H1* (ring finger and CCCH-type domains 1RC3H1, log2FC = −0.86) and *ICE1* (interactor of little elongation complex ELL subunit 1, log2FC = −0.71) (Appendix A). Concerning translation, enrichment analysis identified the overrepresentation of the “ribosome” KEGG pathway (normalized enrichment scores (NES) = 2.60), “translation” Reactome gene set (NES = 2.23), and the “translational elongation” and “translational termination” GO BP (NES = 1.98 and 2.06, respectively) (Appendix A). Enrichment results also highlighted the perturbation in OXPHOS. GSEA identified the significant positive enrichment of “oxidative phosphorylation” in Hallmark (NES = 1.98), KEGG (NES = 1.85), and GO terms (NES = 2.1), in addition to Reactome gene sets linked to OXPHOS as “NADH dehydrogenase complex assembly” (NES = 1.93) or “the citric acid cycle and respiratory electron transport” (NES = 2.22) (Appendix A). In line with these results, enrichment in “mitochondrial fatty acid (FA) β-oxidation” Reactome gene set (NES = 1.83) was also identified. 

#### 3.1.2. In TE, High Insulin Impacted Cellular Energy Metabolism and Oxidative Stress Pathways

Transcriptome analysis by PCA and hierarchical clustering showed no separation between HI TE and CNTRL TE (Figure 2). HI exposure resulted in the differential expression of only one gene, *PNLIP* (pancreatic lipase; log2FC = 5.1) (Figure 3 and Appendix A). However, GSEA analysis identified the significant positive enrichment of 83 gene sets (3 Hallmarks, 5 KEGG pathways, 20 GO BP, and 55 Reactome) and the significant negative enrichment of 7 gene sets (6 Hallmarks and 1 GO BP) (Appendix A). 

Enriched pathways included gene sets implicated in translation and in OXPHOS (Figure 4B). Enrichment results highlighted a few gene sets implicated in translation, such as the “ribosome” KEGG pathway (NES = 1.82) or the “translation” Reactome gene set (NES = 1.82) (Figure 4B and Appendix A). Enrichment results related to OXPHOS included the overrepresentation of the “oxidative phosphorylation” gene sets in Hallmark (NES = 2.39), KEGG (NES = 2.27), and GO BP (NES = 2.11) or the Reactome gene set “respiratory electron transport” (NES = 2.30) (Figure 4B and Appendix A). In addition, reactive oxygen species (ROS) gene set “ROS and RNS production in phagocytes” (NES = 1.86) (Appendix A) and mitochondrial FA β-oxidation Reactome gene set (NES = 1.77) were identified. HI TE also showed the overrepresentation of activated NF-kB-related gene sets, including the Reactome FCERI-mediated NF-kB activation (NES = 1.94) (Appendix A).

#### 3.1.3. High Insulin Induced Common Responses in ICM and TE

Between ICM and TE of high insulin-exposed embryos, whereas no common DEG was observed, 22 shared enriched GSEA gene sets were identified (Appendix A). All shared gene sets exhibited the same level of enrichment. Among shared gene sets we highlighted translation, OXPHOS and FA β-oxidation. Enrichment of ROS and NF-kB signaling was only observed in HI TE. 

### 3.2. Impact of High Glucose In Vitro Exposure

High glucose exposure led to a significant increase in blastocyst rate, mirrored by a significant reduction in the rate of compacted embryos compared to CNTRL embryos (Table 1). No significant differences were observed in the rate of arrested embryos after development with HG (Table 1). Quantification of total cell number showed a significantly increased cell number in HG (263 ± 9, *n* = 75) versus CNTRL (240 ± 7, *n* = 76) blastocysts (*p* < 0.05, Appendix A).

#### 3.2.1. In ICM, High Glucose Altered OXPHOS, Decreased Proliferation, Increased Apoptosis

Transcriptome analysis by PCA and hierarchical clustering showed the separation between HG ICM and CNTRL ICM (Figure 2). Differential analysis showed 41 DEG (24 upregulated and 17 downregulated) in the ICM of embryos exposed to HG compared to the CNTRL ICM (Figure 3 and Appendix A). GSEA analysis identified the significant positive enrichment of 73 functional gene sets (2 Hallmarks, 2 KEGG pathways, 11 GO BP, and 58 Reactome) (Appendix A). 

Enrichment analyses identified 3 main clusters: translation, regulation of the cell number, and OXPHOS (Figure 5A). First, the protein translation cluster included KEGG “ribosome” (NES = 2.67), Reactome “metabolism of amino acids and derivatives” (NES = 1.79) and “translation” (NES = 2.51), and GO BP “translational initiation” (NES = 2.35) gene sets (Figure 5A, Appendix A). In addition, perturbation of transcription was also observed (Appendix A). Genes implicated in transcription were found to be differentially expressed, such as *KDM5A* (lysine demethylase 5A, log2FC = −0.35) and *GATA3* (GATA binding protein 3, log2FC = 1.05) (Appendix A). The second cluster highlighted alterations in the regulation of the cell number. GSEA revealed the enrichment of Hallmark “myc Target v1” (NES = 2.26), Reactome pathways such as “regulation of mitotic cell cycle” (NES = 1.89), and “regulation of apoptosis” (NES = 1.89). The differential analysis identified the overexpression of *LIN54* (lin-54 DREAM MuvB core complex component, log2FC = 0.65) and the underexpression of *CHP2* (calcineurin-like EF-hand protein 2, log2FC = −2.3) and *APC* (APC regulator of WNT signaling pathway, log2FC = −0.67) (Appendix A). To investigate cell proliferation and apoptosis at the cellular level, we performed EdU incorporation and the TUNEL assay (Figure 6 and Appendix A). Indeed, a reduced number of proliferating cells (Figure 6A and Appendix A) and an increased proportion of apoptotic cells (Figure 6B and Appendix A) were identified in HG ICM compared to CNTRL ICM. The third identified cluster concerning perturbations in energy metabolism included Hallmark “oxidative phosphorylation” (NES = 1.70), Reactome “respiratory electron transport” (NES = 2.23), GO BP “mitochondrial respiratory chain complex assembly” (NES = 2.02), and Reactome “cellular response to hypoxia” (NES = 1.81) (Figure 5A and Appendix A). In line with these results, HG ICM showed the overexpression of *FH* (fumarate hydratase, log2FC = 0.45) compared to CNTRL ICM (Appendix A). 

In addition to the main clusters, HG ICM transcriptomes showed perturbations in signaling pathways. HG ICM showed the overexpression of *REL* (REL proto-oncogene, NF-kB subunit, log2FC = 2.18) and the enrichment of gene sets related to NF-kB signaling, such as Reactome “FCERI mediated NF-kB activation” (NES = 1.84) (Appendix A). Differential and enrichment analyses also showed dysregulations in the WNT signaling pathway. These results included the downregulation of *APC* (log2FC = −0.67) (Appendix A) and enrichment of Reactome “degradation of β-catenin by the destruction complex” (NES = 1.73) (Appendix A). Furthermore, gene-by-gene analysis of the HG ICM DEG revealed the overexpression of genes involved in the trophoblast lineage, such as GATA binding protein 3 (*GATA3*, log2FC = 1.05) and placenta expressed transcript 1 (*PLET1*, log2FC = 2.39) (Figure 7 and Appendix A). 

#### 3.2.2. In TE, High Glucose Impacted Metabolic Pathways, Increased Proliferation, and Decreased Apoptosis

Transcriptome analysis by PCA and hierarchical clustering did not show a separation between HG TE and CNTRL TE (Figure 2). HG TE showed 132 DEG compared to CNTRL TE (53 overexpressed and 79 underexpressed) (Figure 3 and Appendix A). GSEA identified the enrichment of 78 functional gene sets, 76 of which were positively enriched (10 Hallmarks, 1 KEGG, and 65 Reactome), and 2 (GO BP) were negatively enriched (Appendix A). 

Enriched pathways identified clusters related to metabolism and cell number regulation (Figure 5B). Concerning alterations in metabolism, enrichment results highlighted perturbations in mTOR signaling by the overrepresentation of Hallmarks “mTORC1 signaling” (NES = 1.68) and “PI3K-AKT-mTOR signaling” (NES = 1.61) (Figure 5B, Appendix A). Alteration of numerous genes involved in glycolysis, glycine, and lipid metabolism were identified such as *HK1* (hexokinase 1, log2FC = 0.53), *PHGDH* (phosphoglycerate dehydrogenase, log2FC = 0.85), *AACS* (acetoacetyl-CoA synthetase, log2FC = −1.48), *LDLR* (low density lipoprotein receptor, log2FC = −1.40, *GPAT3* (glycerol-3-phosphate acyltransferase 3, log2FC = 1.22), and *GPCPD1* (glycerophosphocholine phosphodiesterase 1, log2FC = 0.88) (Appendix A). The second main enrichment concerned the regulation of cell number (Figure 5B). Enriched pathways included Hallmark “myc Target v1” (NES = 2.01), Reactome “mitotic G1-G1/S phases” (NES = 2.02), and “regulation of mitotic cell cycle” (NES = 2.08) gene sets (Appendix A). Consistent with these results, the HG TE transcriptome showed the altered expression of cell cycle progression genes, such as the underexpression of *CDKL4* (cyclin dependent kinase like 4, log2FC = −1.82) or *SMARCD3* (SWI/SNF related matrix associated actin dependent regulator of chromatin subfamily d member 3, log2FC = −3.00) or the overexpression of *GADD45A* (growth arrest and DNA damage inducible alpha, log2FC = 0.85) or *WEE1* (WEE1 G2 Checkpoint Kinase, log2FC = 0.97) (Appendix A). Assessment of cell proliferation by the EdU incorporation assay in HG TE further confirmed an increased number of proliferating cells compared to CNTRL TE (Figure 6C and Appendix A). In parallel, enrichment results showed the overrepresentation of apoptosis-related gene sets such as Hallmarks “apoptosis” (NES = 1.73), “P53 pathway” (NES = 1.75), and the Reactome “regulation of apoptosis” (NES = 1.77) gene set (Appendix A). HG TE exhibited the differential expression of genes implicated in apoptosis, such as *CASP7* (caspase 7; log2FC = 1.13), *PDCD6* (programmed cell death 6; log2FC = 0.59), and *TRADD* (TNFRSF1A associated via death domain; log2FC = −1.37) (Appendix A). Investigation of apoptosis in HG TE by TUNEL assay showed a decrease in the number of apoptotic cells compared to CNTRL TE (Figure 6D and Appendix A). 

Beyond these main clusters, HG TE transcriptomes exhibited several other perturbations, as in the immune response. Enrichment and differential analysis identified the Hallmark “TGF-β signalling” (NES = 1.64) and “TNF-α signalling via NF-kB” (NES = 1.52) gene sets, and the underexpression of *ERC1* (ELKS/RAB6-interacting/CAST family member 1, log2FC = −0.91) (Appendix A). The WNT signaling was identified as deregulated as highlighted by the enrichment of the Reactome “degradation of β-catenin by the destruction complex” (NES = 1.81) and “β-catenin independent WNT signaling” (NES = 1.79), and the underexpression of *ANKRD10* (ankyrin repeat domain 10; log2FC = −1.13) (Appendix A). In addition, HG TE showed the enrichment of gene sets implicated in transcription and translation, such as the Reactome “transcriptional activity of SMAD2 SMAD3:SMAD4 heterotrimer” (NES = 1.9) and “translation” (NES = 1.88) (Appendix A). Along with these gene sets, gene-by-gene analysis and functional annotation by DAVID showed several DEG associated with transcriptional regulation, chromatin remodeling, and epigenetic mechanisms (Figure 8 and Appendix A). Among the HG TE DEG, we highlighted the overexpression of *GADD45A*, *WEE1*, and *NPM3* (nucleophosmin/nucleoplasmin 3), and the underexpression of *SMARCD3*, *PADI2* (peptidyl arginine deiminase 2), *MOV10L1*, *ATF7* (activating transcription factor 7), *RESF1* (retroelement silencing factor 1), *BPTF* (bromodomain PHD finger transcription factor), and *NSD3* (nuclear receptor binding SET domain protein 3) (Figure 8 and Appendix A).

#### 3.2.3. High Glucose Induced Common and Specific Responses in ICM and TE

From the DEG identified in HG ICM (*n* = 41) and HG TE (*n* = 132), only 3 were shared: *ARRDC4* (arrestin domain containing 4) (log2FC = 2.07 and 2.02, respectively), *FAM3D* (family with sequence similarity 3 member D) (log2FC = 0.96 and 0.54, respectively), and *MOV10L1* (Mov10 RISC complex RNA helicase like 1) (log2FC = −1.26 and −1.57, respectively) (Appendix A). Despite the small amount of shared DEG, several processes were common, as shown by the 37 shared gene sets identified by GSEA, which are overrepresented in both ICM and TE (Appendix A). These gene sets were mainly related to the regulation of cell number. However, we identified opposite responses: the HG ICM exhibited decreased proliferation and increased apoptosis, whereas the HG TE showed increased proliferation and reduced apoptosis. Specific responses included transcriptome changes related to OXPHOS and lineage commitment in HG ICM and metabolism and epigenetic regulation in HG TE. 

### 3.3. Impact of High Glucose and High Insulin In Vitro Exposure

High glucose and high insulin exposure led to a significant increase in blastocyst rate, mirrored by a significant reduction in the rate of compacted embryos compared to CNTRL embryos (Table 1). No significant differences were observed in the rate of arrested embryos after development with HGI (Table 1). Quantification of total cell number showed a significantly increased cell number in HGI (285 ± 14, *n* = 50) versus CNTRL (240 ± 7, *n* = 76) blastocysts (*p* < 0.05, Appendix A).

As the development of HI embryos was not different from that of CNTRL embryos, a comparison of HGI vs. HI resulted in similar observations to HGI vs. CNTRL: no difference in the rate of arrested embryos, decrease in the rate of compacted embryos, and an increase of the rate of blastocysts and blastocyst total cell number (Table 1 and Appendix A).

In comparison to high glucose, HGI embryos displayed similar development parameters: the rates of arrested, compacted, and blastocyst were similar in HGI compared to HG embryos. The cell number was also similar in HGI blastocysts in comparison to HG blastocysts (Table 1 and Appendix A). 

#### 3.3.1. In ICM, Alteration of OXPHOS and ROS by High Glucose and High Insulin

Transcriptome analysis by PCA showed the separation between HGI ICM and CNTRL ICM (Figure 2). Differential analysis showed 39 DEG (20 overexpressed and 19 underexpressed) in comparison to CNTRL ICM (Figure 3 and Appendix A). GSEA analysis showed the significant positive enrichment of 107 gene sets (5 Hallmarks, 7 KEGG, 28 GO BP, and 66 Reactome) (Appendix A). 

Enriched pathways highlighted two main clusters related to the regulation of gene expression and cellular energy metabolism (Figure 9A). Firstly, concerning the regulation of gene expression, several transcription factors were differentially expressed, such as *ELF2* (E74 like ETS transcription factor 2; log2FC = 0.71) and *SREBF2* (sterol regulatory element binding transcription factor 2; log2FC = −0.49). Overexpression of genes implicated in mRNA processing, such as *PTBP2* (polypyrimidine tract binding protein 2, log2FC = 0.96), was observed (Appendix A). Coherent with this, enrichment results highlighted transcriptome changes related to translation, including GO BP “translational initiation” (NES = 2.33), Reactome “mRNA splicing” (NES = 1.74), and KEGG “ribosome” (NES = 2.85) (Figure 9A and Appendix A). The second identified cluster was related to OXPHOS. HGI ICM showed transcriptomic changes including Hallmark “oxidative phosphorylation” (NES = 2.02), KEGG (NES = 2.26), GO BP (NES = 2.55), and Reactome “mitochondrial fatty acid β oxidation” (NES = 1.82) gene sets (Figure 9A and Appendix A). Additionally, HG ICM showed the enrichment of ROS pathways, including Reactome “cellular response to hypoxia” (NES = 1.78) and Hallmark “reactive oxygen species” (NES = 1.73) (Figure 9A and Appendix A). 

In addition to these main clusters, HGI ICM exhibited DEG and enriched gene sets implicated in cell number regulation. Among these, we can list the *APC* gene (log2FC = −0.74), the Hallmark “myc target v1” (NES = 2.02), and the Reactome “regulation of apoptosis” (NES = 1.73) (Appendix A). We examined a possible imbalance in proliferation and apoptosis in HGI ICM and did not detect significant changes in the proportion of proliferating or apoptotic cells compared to CNTRL ICM (Figure 6A,B and Appendix A). HGI ICM transcriptomes also showed enrichment in TNF-α signaling (Figure 9A and Appendix A). Among the DEG and gene sets implicated in this pathway, we highlighted the differential expression of *USP15* (ubiquitin specific peptidase 15, log2FC = 0.47), enriched Hallmark “TNFA signaling via NFKB” (NES = 1.70), and GO BP “cytokine metabolic process” (NES = 1.94) (Figure 9A and Appendix A). Among the DEG identified, we also highlighted the differential expression of genes implicated in the trophoblast lineage and placenta development, such as *PLET1* (log2FC = 1.97) and *PEG10* (paternally expressed 10, log2FC = 1.79) (Figure 7 and Appendix A). 

Then, the specific responses triggered by high glucose and high insulin in combination on the ICM were determined by the identification of common and specific transcriptomic changes between HGI versus CNTRL ICM and HI versus CNTRL ICM and between HGI versus CNTRL ICM and HG versus CNTRL ICM (Appendix A). 

HGI ICM and HI ICM shared 2 DEG, including the protein-coding gene *RPS6KA3* (ribosomal protein S6 kinase A3) (log2FC = 0.77 and 0.88, respectively), and 29 gene sets, all with the same expression pattern (Appendix A). Shared changes in gene expression were related to the regulation of gene expression and OXPHOS (Appendix A). 

HGI ICM and HG ICM shared 16 DEG and 62 gene sets, all with the same expression pattern (Appendix A). These shared transcriptome changes were related to the regulation of gene expression, OXPHOS, NF-kB signalling, and the aberrant expression of TE genes. Despite the few shared gene sets involved in cell number regulation, the increased apoptosis and decreased proliferation identified in HG ICM were not found in HGI ICM. The enrichment in the ROS pathway was exclusively identified in HGI ICM.

#### 3.3.2. In TE, Alteration of OXPHOS, ROS, and Proliferation by High Glucose and High Insulin

Transcriptome analysis by PCA and hierarchical clustering did not show a separation between HGI TE and CNTRL TE (Figure 2C). HGI TE exhibited 16 DEG (10 overexpressed and 6 underexpressed) in comparison to CNTRL TE (Figure 3 and Appendix A). Enrichment analysis in HGI TE identified 108 gene sets positively enriched (8 Hallmarks, 2 KEGG pathways, 11 GO BP, and 87 Reactome) (Appendix A). 

Enriched pathways included gene sets related to transcription, translation, cell number regulation, and OXPHOS, as highlighted by the three main clusters (Figure 9B and Appendix A). Transcription and translation enriched gene sets included the Reactome “transcriptional regulation by MECP2” (NES = 1.89), KEGG “ribosome” (NES = 2.53), Reactome “translation” (NES = 2.10), and KEGG “proteasome” (NES = 2.17) (Appendix A). Enriched pathways implicated in the regulation of cell number included Hallmark “myc target v1” (NES = 2.30) and Reactome “regulation of mitotic cell cycle” (NES = 2.22) (Figure 9B and Appendix A). In line with these enriched gene sets, the differential analysis identified the significant overexpression of *TUBB* (tubulin beta class I, log2FC = 0.50) and *WEE1* (log2FC = 0.92), 2 genes implicated in the G2/M transition of the mitotic cell cycle (Appendix A). The EdU incorporation assay in HGI TE confirmed a significant increase in the proportion of proliferating cells compared to CNTRL TE (Figure 6C and Appendix A). Despite identifying a small enrichment of gene sets implicated in apoptosis (Appendix A), the TUNEL assay in HGI TE did not detect significant differences in the proportion of apoptotic cells compared to CNTRL TE (Figure 6D and Appendix A). The third identified cluster was related to energy metabolism, as indicated by the Hallmark “oxidative phosphorylation” (NES = 1.54), “ROS pathway” (NES = 1.79), Reactome “cellular response to hypoxia” (NES = 2.00), and GO BP “regulation of transcription from RNA polymerase II promoter in response to hypoxia” (NES = 2.04) (Figure 9B and Appendix A). In line with these enrichments, DEG in HGI TE included the overexpression of *TXNIP* (thioredoxin-interacting protein; log2FC = 1.81) (Appendix A). 

Besides these clusters, transcriptome analysis showed perturbations in the immune response in HGI TE. Enriched pathways included the Hallmark “TGF-α signaling via NF-kB” (NES = 1.73), GO BP “positive regulation of cytokine biosynthetic process” (NES = 2.02), and TGF-β signaling (NES = 1.67) (Appendix A). Additionally, enrichment results showed perturbations of the WNT signaling pathway by the Reactome “degradation of β-catenin by the destruction complex” (NES = 1.80) (Appendix A). 

The combination of high glucose and high insulin triggered specific responses in the TE. HI TE and HGI TE shared 1 DEG (*PNLIP*, log2FC = 5.08 and 4.71, respectively) and 43 overrepresented gene sets (Appendix A). These gene sets were related to translation, OXPHOS, ROS, and NF-kB signaling (Appendix A). In comparison to HG TE, 9 DEG and 53 gene sets shared the same expression pattern (Appendix A). Corresponding pathways were related to transcription and translation, NF-kB signaling and regulation of cell number. A higher number of proliferating cells was detected in both HG TE and HGI TE. Decrease in HG TE, apoptosis was not altered in HGI TE. The alteration of the metabolic pathway genes and the altered expression of genes involved in epigenetic mechanisms detected in HG TE were not identified in HGI TE. Inversely, OXPHOS and ROS pathways were only overrepresented in HGI TE (Appendix A).

#### 3.3.3. High Glucose and High Insulin Induced Common and Specific Responses in ICM and TE

Between HGI ICM and HGI TE, whereas only one DEG was shared (*ARRDC4*, log2FC = 2.09 and 2.02, respectively), half (*n* = 54) of the enriched gene sets were shared and exhibited the same enrichment pattern (Appendix A). Transcription, translation, OXPHOS, ROS, and NF-kB signaling were impacted in ICM and TE in response to HGI (Appendix A). Specific responses between compartments can be noticed, such as cell commitment dysregulations occurring exclusively in the ICM. 

## 4. Discussion 

Prediabetes and the early stages of T2D are characterized by hyperglycemia and hyperinsulinemia [1,2]. Unfortunately, these first metabolic dysregulations are often asymptomatic, resulting in nearly half of people with T2D being undiagnosed and untreated [1]. In the early stages of pregnancy, including the preimplantation period, women are not yet aware of their gestational status; therefore, in undiagnosed diabetic women, pregnancies are not adequately intervened. Increased glucose and insulin concentrations are reflected in oviductal and uterine fluids [9,11]. Preimplantation embryos are responsive to glucose and insulin through the activation of signaling and metabolic pathways [14,15,41,42,43]. The preimplantation period corresponds to a critical window of susceptibility during which variations in the environment can have a major impact on the offspring. Here, we have established an in vitro model using the rabbit to study the effects of high glucose and/or high insulin on preimplantation embryo development. 

As growth factors, glucose and insulin are key regulators of proliferation and apoptosis. In the present study, the presence of high glucose stimulated blastocyst development and growth. Observations in mice and bovine embryos mainly described a negative impact of glucose on blastocyst development, obtained with glucose concentrations above 20 mM [11,44]. Here, we have shown that high glucose exposure led to the alteration of proliferation and apoptosis in mirror patterns. Consistent with our findings, mouse and rat embryos exposed to glucose showed increased apoptosis in the ICM [45,46]. Inversely to ICM, and to our knowledge first described here, proliferation was increased, and apoptosis decreased in the TE of embryos exposed to high glucose. When embryos were exposed to high levels of insulin alone, blastocyst rate and growth were not impacted, and changes in the expression of genes involved in proliferation and apoptosis were not identified. The mitogenic actions of insulin are well known [18]; however, in preimplantation embryos, this remains controversial [47,48,49,50]. Our findings show that when high levels of insulin were added in addition to high glucose, the increased rate and growth of blastocysts observed in the presence of high glucose alone persisted. Despite changes in the proliferation and apoptosis gene expression, only the proliferation rate remained increased in the TE. These results suggest a crosstalk between glucose and insulin in mediating growth-related effects. 

Glucose and insulin play a central role in regulating energy homeostasis and metabolism [18,51]. Here, embryos developed in the presence of high glucose exhibited OXPHOS signatures in the ICM and mTORC1 signaling and glycolytic and lipid metabolism signatures in the TE. Glucose, via glycolysis and OXPHOS, leads to the production of cellular energy in the form of ATP [52]. Until the morula stage, preimplantation embryos metabolize lactate and pyruvate preferentially as an energy source through OXPHOS [51]. Around the morula stage and onward, glucose is preferentially metabolized, although the metabolic pathway used may differ between the ICM and TE [51,53]. Exposure to hyperglycemia in vitro and in vivo led to hyperactivation of mTORC1 signaling in rabbit blastocysts, especially in the TE [41]. The mTORC1 and mTORC2 complexes stimulate anabolic processes such as protein, lipid, and nucleotide synthesis and regulate glucose metabolism by favoring glycolysis over OXPHOS [54]. To sustain cell growth, the mTORC1 and mTORC2 complexes stimulate anabolic processes such as protein, lipid, and nucleotide synthesis, regulate glucose metabolism by favoring glycolysis over OXPHOS, and promote cell survival and proliferation [54]. In the present study, both ICM and TE developed in an insulin-rich environment and exhibited OXPHOS gene expression signatures. The metabolic effects of insulin are well known, and insulin has been shown to stimulate the oxidative capacity of mitochondria [18,55]. In addition, the TE of embryos developed with high insulin showed ROS-related gene expression changes, suggesting insulin-mediated oxidative stress. ROS, mainly produced as a by-product of OXPHOS, plays a role in physiological cellular processes [56]. Here, in the presence of both high glucose and high insulin, OXPHOS and ROS gene expression signatures were also identified in the ICM and TE. In addition, transcriptome changes related to NF-kB and TNF-α signaling were identified in the ICM and TE. NF-kB signaling, central regulator of inflammation and immunity, also regulates multiple cellular processes, including mitochondrial respiration [57]. NF-kB is induced by environmental cues, including insulin and ROS [58,59]. Here, gene expression changes related to OXPHOS, ROS, and NF-kB suggest metabolic stress in the ICM and TE of embryos exposed to both high glucose and high insulin. 

Interestingly, we identified the deregulation of a subset of genes implicated in chromatin remodeling and epigenetic regulation in the TE. Emerging research has underlined the crosslink between metabolism and chromatin dynamics and its influence on gene expression [60,61]. A clear example of this is the generation of regulators of chromatin-modifying enzymes through glucose metabolism [60,61]. Among the genes showing altered expression in the TE exposed to high glucose, we highlighted *GADD45A*, *BPTF*, *PADI2*, and *ATF7*. GADD45A mediates active DNA demethylation, facilitating transcriptional activation, and also regulates trophoblast cell migration and invasion during placentation [62,63]. BPTF encodes the largest subunit of the Nucleosome Remodeling Factor (NURF) chromatin remodeling complex and plays an essential role in extraembryonic lineage development [64,65]. As for PADI2, a catalyzer of histone citrullination, it regulates chromatin organization and transcriptional regulation of cell cycle progression, metabolism, and proliferation genes [66]. ATF7, a stress-responsive chromatin regulator that recruits histone methyltransferases to repress the transcription of metabolic genes, has been proposed to mediate paternal low protein diet-induced intergenerational programming by reducing H3K9me2 in target genes [67,68]. In the presence of both high glucose and high insulin, the number of epigenetic genes with altered expression was less than in embryos exposed to high glucose alone, whereas no gene with an epigenetic-related function showed differential expression in embryos exposed to high insulin alone. Thus, these results suggested a crosstalk between insulin and glucose in terms of epigenetic regulation, especially in the TE. Thus, the differential expression of these genes suggests alterations in the TE epigenetic landscape, alterations that could compromise trophoblast differentiation. 

In addition to the altered expression of epigenetic genes in the TE, the ICM exhibited the overexpression of genes involved in the trophoblast lineage when exposed to high glucose alone or in combination with high insulin. GATA3 is a well-known transcription factor associated with TE initiation and trophoblast differentiation [69,70]. Overexpression of GATA3 was sufficient to induce trophoblast fate in mouse embryonic stem cells (ESCs) [69]. PLET1 is an epigenetically-regulated cell surface protein essential to drive the differentiation of the trophoblast lineage [71]. *PEG10*, a paternally expressed imprinted gene highly expressed in the placenta, is essential for placenta formation in early development [72]. In the mouse, it has been recently demonstrated that glucose metabolism is required for the specification of the TE lineage through the hexosamine biosynthetic pathway (HBP), the pentose phosphate pathway (PPP), and the activation of the mTOR pathway [70]. Furthermore, when human ESCs were cultured with high glucose, the differentiation of the definitive endoderm was impaired [73]. In addition, we observed the enrichment of NF-kB signatures on the ICM of embryos exposed to high glucose alone or in combination with high insulin, and the NF-kB signaling pathway is known to regulate trophoblast differentiation and function [74,75]. Moreover, the ICM of embryos exposed to high glucose alone or in combination with high insulin showed signatures of oxidative rather than glycolytic metabolism, which in the mouse has been described to be characteristic of the TE rather than of the ICM [20]. Our findings indicate a potential impairment in cell commitment in the ICM. Perturbations in ICM cell allocation could directly influence the TE lineage [46,76]. Blastocysts with different amounts of ICM cells led to limited trophoblast proliferation, suggesting the necessity for cell allocation homeostasis between these two compartments [46,76]. Moreover, the crosstalk between ICM and TE influencing TE differentiation has been previously described [17]. 

In conclusion, exposure to high glucose and high insulin alone or in combination during preimplantation development results in lineage-specific responses in the progenitors of the future individual and the embryonic portion of the placenta. We showed here that in the presence of high insulin, the impact of high glucose was lowered in some cases, suggesting significant crosstalk between glucose metabolism and insulin signaling in the early embryo. These results suggested that a mismatch in the glucose and insulin axis represents a risk for early embryonic development and, thus, for offspring health. Moreover, despite being present in the preimplantation maternal environment, insulin is usually absent in in vitro culture systems. Integration of insulin may be useful in improving embryo culture media. 

## Figures and Tables

**Figure 1 cells-11-03766-f001:**
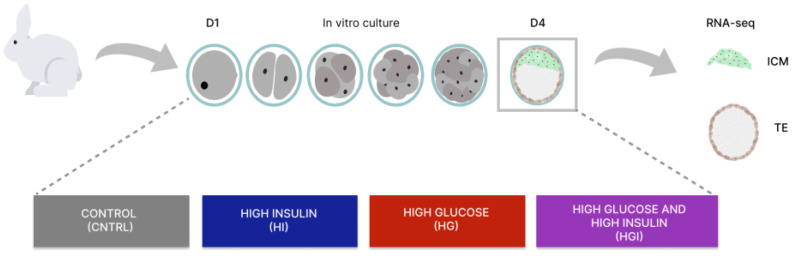
Schematic representation of the experimental workflow to analyze the in vitro exposure of preimplantation embryos from 1-cell to blastocyst stage for control, high insulin, high glucose, and high glucose and high insulin. The inner cell mass (ICM) and trophectoderm (TE) transcriptomes were determined by RNA-seq. D1, day 1. D4, day 4.

**Figure 2 cells-11-03766-f002:**
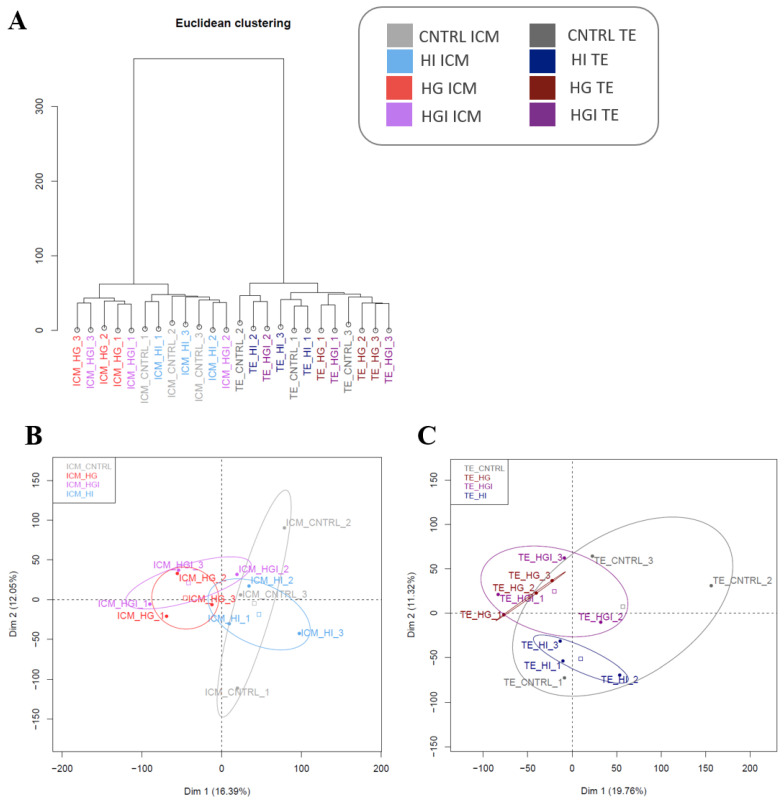
Transcriptome analysis of isolated ICM and TE from in vitro-developed blastocysts with high glucose and/or high insulin. (**A**). Clustering by Euclidean distance of the transcriptomic datasets of ICM and their corresponding TE developed in CNTRL, HI, HG, or HGI. Each group included three biological replicates which consisted of *n* = 11-16 ICM or TE. (**B**). Principal component analysis (PCA) of ICM groups. (**C**). PCA of TE groups. ICM, inner cell mass. TE, trophectoderm. CNTRL, control; HI, high insulin; HG, high glucose; HGI, high glucose and high insulin. Samples are color-coded according to the legend at the top (right).

**Figure 3 cells-11-03766-f003:**
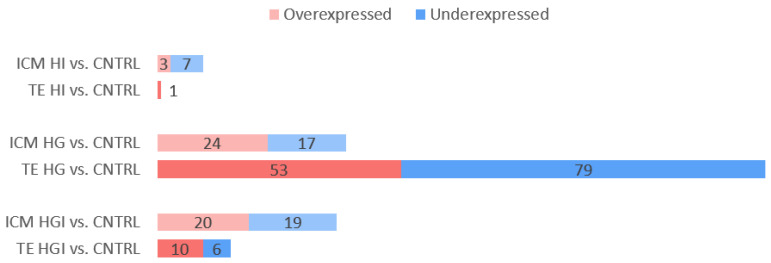
Differentially expressed genes (DEG) in ICM and TE of in vitro-developed blastocysts with HI, HG, or HGI compared to CNTRL. The number of overexpressed (red) and underexpressed (blue) DEGs with *p*-adjusted < 0.05 are shown. ICM, inner cell mass. TE, trophectoderm.

**Figure 4 cells-11-03766-f004:**
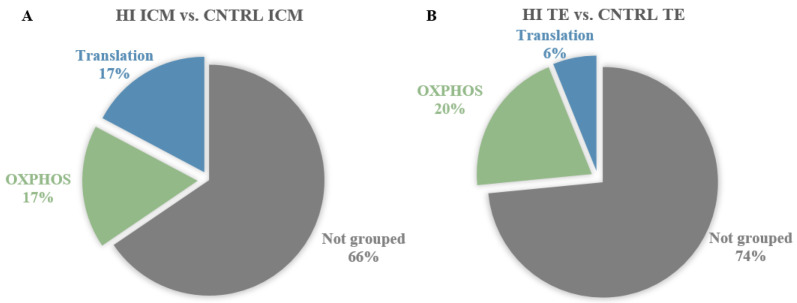
Significantly enriched gene sets (FDR < 0.05) in ICM and TE transcriptomes of in vitro-developed blastocysts with HI compared to CNTRL. Significantly enriched gene sets were identified by GSEA with the Molecular Signature Database (MSigDB) gene set collections: Hallmarks, KEGG, Reactome, and GO BP. GSEA was followed by SUMER analysis for gene set condensation. (**A**). Pie charts showing the enriched gene sets in HI ICM versus CNTRL ICM. (**B**). Pie charts showing the enriched gene sets in HI TE versus CNTRL TE. ICM, inner cell mass. TE, trophectoderm.

**Figure 5 cells-11-03766-f005:**
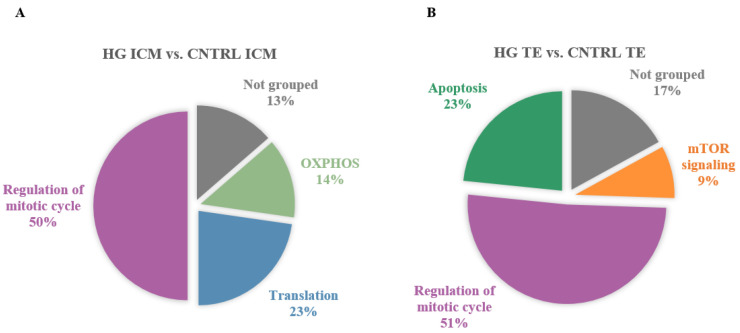
Significantly enriched gene sets (FDR < 0.05) in ICM and TE transcriptomes of in vitro-developed blastocysts with HG compared to CNTRL. Significantly enriched gene sets were identified by GSEA with the Molecular Signature Database (MSigDB) gene set collections: Hallmarks, KEGG, Reactome, and GO BP. GSEA was followed by SUMER analysis for gene set condensation. (**A**). Pie charts showing the enriched gene sets in HG ICM versus CNTRL ICM. (**B**). Pie charts showing the enriched gene sets in HG TE versus CNTRL TE. ICM, inner cell mass. TE, trophectoderm.

**Figure 6 cells-11-03766-f006:**
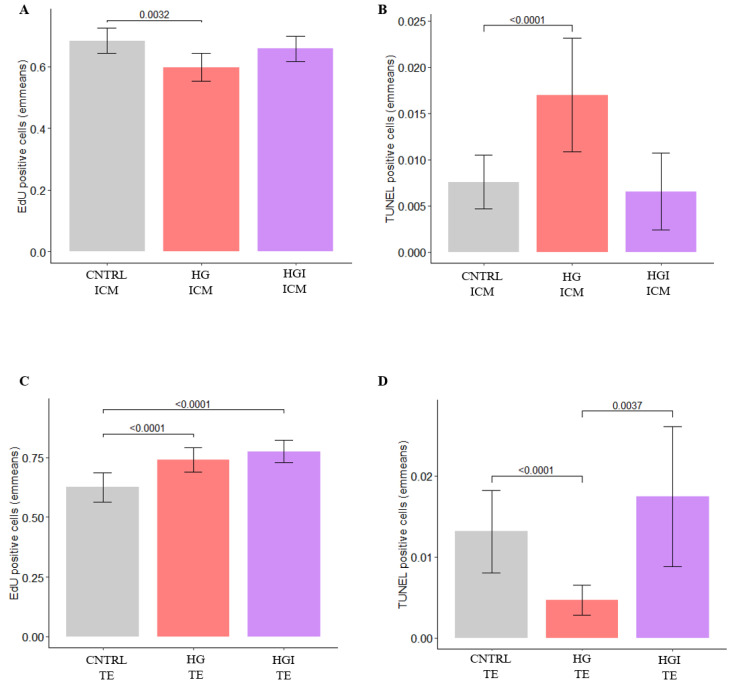
Quantification of proliferating and apoptotic cells in the ICM and TE of in-vitro-developed blastocysts with CNTRL, HG, and HGI by EdU incorporation and TUNEL assays. (**A**) Barplots showing the emmeans of proliferating cells in the ICM (n ICM = 16–38). (**B**) Barplots showing the emmeans percentage of apoptotic cells in the ICM (n ICM = 13–59). (**C**) Barplots showing the emmeans of proliferating cells in the TE (n TE = 16–24). (**D**) Barplots showing the emmeans of apoptotic cells in the TE (n TE = 18–52). Values are presented as emmeans ± S.E. Significant *p* values (*p* < 0.05) are shown. ICM, inner cell mass. TE, trophectoderm; CNTRL, control; HG, high glucose; HGI, high glucose and high insulin.

**Figure 7 cells-11-03766-f007:**
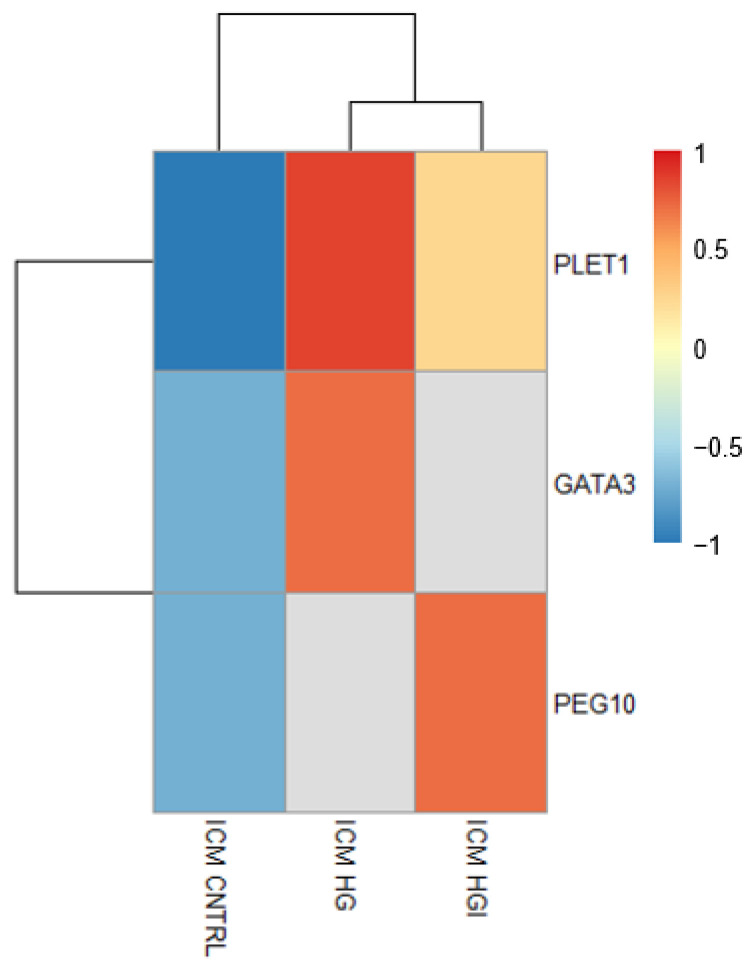
Heatmap showing the differential expression of genes (DEG) implicated in the TE lineage in HG and HGI ICM compared to CNTRL ICM. The mean normalized expression counts of n = 3 biological replicates, transformed to a Z-score, are represented by the color key. The gray color indicates the gene is not a DEG in that group. ICM, inner cell mass. TE, trophectoderm.

**Figure 8 cells-11-03766-f008:**
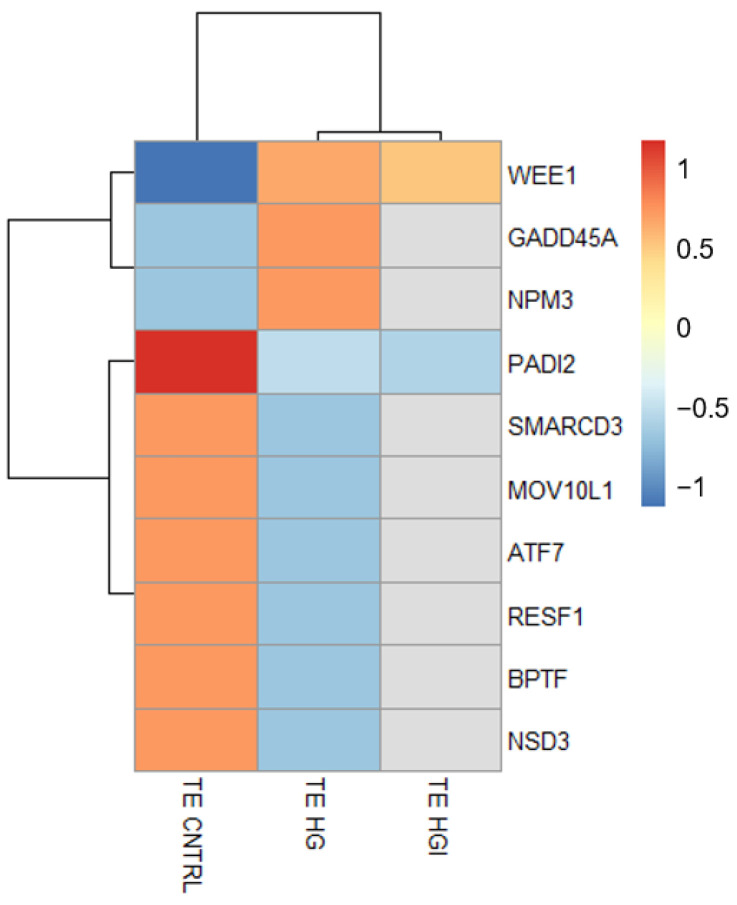
Heatmap showing the differential expression of genes (DEG) with a role in epigenetic regulation in HG and HGI TE compared to CNTRL TE. The mean normalized expression counts of *n* = 3 biological replicates, transformed to a z-score, are represented by the color key. The gray color indicates that the gene is not a DEG in that group. ICM, inner cell mass. TE, trophectoderm.

**Figure 9 cells-11-03766-f009:**
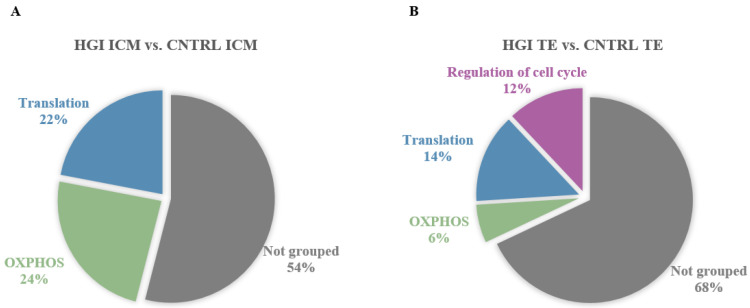
Significantly enriched gene sets (FDR < 0.05) in the ICM and TE transcriptomes of in vitro-developed blastocysts with HGI compared to CNTRL. Significantly enriched gene sets were identified by GSEA with the Molecular Signature Database (MSigDB) gene set collections: Hallmarks, KEGG, Reactome, and GO BP. GSEA was followed by SUMER analysis for gene set condensation. (**A**). Pie charts showing the enriched gene sets in HGI ICM versus CNTRL ICM. (**B**). Pie charts showing the enriched gene sets in HGI TE versus CNTRL TE. ICM, inner cell mass. TE, trophectoderm.

**Table 1 cells-11-03766-t001:** Developmental competence of rabbit preimplantation embryos developed in vitro in CNTRL, HI, HG, or HGI conditions. Values are expressed as emmeans with standard errors in parenthesis. Different superscript letters (a, b) indicate significant differences within the same column (*p* < 0.05). CNTRL, control; HI, high insulin; HG, high glucose; HGI, high glucose and high insulin.

Condition	*N* Rabbits	*N* Embryos	Development Arrest Rate	Compacted Embryos Rate	Blastocyst Rate
CNTRL	60	1090	0.034 (0.009) ^a^	0.303 (0.061) ^a^	0.638 (0.057) ^a^
HI	21	530	0.029 (0.009) ^a^	0.309 (0.063) ^a^	0.645 (0.059) ^a^
HG	52	751	0.027 (0.008) ^a^	0.228 (0.052) ^b^	0.726 (0.051) ^b^
HGI	35	519	0.023 (0.007) ^a^	0.232 (0.053) ^b^	0.732 (0.051) ^b^

## Data Availability

The data are available under accession number GSE218009GEO.

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
