# Peer review of "Identification of the Inner Cell Mass and the Trophectoderm Responses after an In Vitro Exposure to Glucose and Insulin during the Preimplantation Period in the Rabbit Embryo"

_cells, 2022, doi:10.3390/cells11233766_

Round 1
Reviewer 1 Report
The authors of this paper investigated the effect of high glucose levels and/or hyperinsulinemic environment on the development of the preimplantation embryo in rabbits. I strongly advise them to revise and resubmit their study after making the changes suggested in the following comments.
First, all abbreviations should be fully described in their first mention ... ex, ICM, TE, DOHaD,... etc.
Generally, the abstract should be rewritten. It is not informative and does not reflect the aim of the research.
Please add p value in the abstract when mentioned the results.
The introduction should be revised or rewritten because the majority of the information presented pertains to type 2 diabetes. However, the study and experimental design involving embryos exposed to hyperglycemia or hyperinsulinemia should focus on type 1 diabetes or hyperglycemic condition. The experimental strategies design and data reported are very poor and not very convincing. It is difficult to understand exactly what type of diabetic model they used in this study?
How many repeat experiments have been performed? At least one repeat experiment is required and has to be shown, this is not clear in materials and methods??.
In the results figure 6 why no data about proliferating and apoptotic cells in the ICM and TE in case of high insulin condition, please clarify?
The results should be accompanied by representative images for fluorescence microscopy, DAPI staining, and TUNEL apoptosis assay.
The authors must explain how they selected these glucose and insulin doses for the current experiment.
Title should be very clear in terms of in vitro experiment and abbreviation should be avoided in the title.
Please correct the manuscript for language and grammatical mistakes.
Author Response
Responses to Reviewer #1:
The authors of this paper investigated the effect of high glucose levels and/or hyperinsulinemic environment on the development of the preimplantation embryo in rabbits. I strongly advise them to revise and resubmit their study after making the changes suggested in the following comments.
We thank the Reviewer 1 for reviewing our article and providing us with valuable comments and suggestions. We followed the recommendations to change the manuscript. Here is the point-by-point response:
First, all abbreviations should be fully described in their first mention ... ex, ICM, TE, DOHaD,... etc.
All the abbreviations are now fully described in their first mention.
Generally, the abstract should be rewritten. It is not informative and does not reflect the aim of the research. Please add p value in the abstract when mentioned the results.
Thank you for this suggestion. Abstract has now been rewritten and the results are presented more accurately.
The introduction should be revised or rewritten because the majority of the information presented pertains to type 2 diabetes. However, the study and experimental design involving embryos exposed to hyperglycemia or hyperinsulinemia should focus on type 1 diabetes or hyperglycemic condition. The experimental strategies design and data reported are very poor and not very convincing. It is difficult to understand exactly what type of diabetic model they used in this study?
Thank you for this comment. We added information on T1D in the introduction.
“Type 1 diabetes (T1D), an immune-related disease characterized by the destruction of insulin producing cells, affects a young population (doi: 10.1016/S0140-6736(18)31320-5). In T1D, the glucose-insulin axis is disrupted. Insulin is no longer produced, insulin-stimulated glucose uptake is reduced, resulting in persistent hyperglycemia (doi: 10.1016/S0140-6736(18)31320-5)”.
How many repeat experiments have been performed? At least one repeat experiment is required and has to be shown, this is not clear in materials and methods?
We apologize for the missing information. The following precisions have been added in Materials and Methods sections:
Embryo in-vitro development: “The rate of arrested embryos (developmental arrest), compacted embryos, and blastocysts/cavitated embryos reported in percentage was calculated in fifteen to twenty-nine independent experiments from the total of one-cell embryos placed in culture”.
RNA sequencing: “Total RNA was extracted from three biological replicates per culture condition corresponding of pooled samples (n = 11-16 ICM or TE per replicate)”
Total cell number quantification: “Each condition was analyzed in between five to nine independent experiments”.
TUNEL assay: “Detection of apoptotic cells was performed using the DeadEnd Fluorometric TUNEL System (Promega) in two to six independent experiments”.
EdU incorporation assay: “Detection of proliferating cells was performed using the Click-iT® Edu Imaging kit (Fisher Scientific) in at least three independent experiments”.
In addition, exact number of ICM and TE counted for quantification of proliferation and apoptotis has been added in the legend of Figure 6:
“Quantification of proliferating and apoptotic cells in the ICM and TE of in-vitro-developed blastocysts with CNTRL, HG and HGI by EdU incorporation and TUNEL assays. A. Barplots showing the emmeans of proliferating cells in the ICM (nICM=16-38). B. Barplots showing the emmeans percentage of apoptotic cells in the ICM (nICM=13-59). C. Barplots showing the emmeans of proliferating cells in the TE (nTE=16-24). D. Barplots showing the emmeans of apoptotic cells in the TE (nTE=18-52). Values are presented as emmeans ± S.E. Significant P values (P<0.05) are shown. ICM, inner cell mass. TE, trophectoderm; CNTRL, control; HG, high glucose; HGI, high glucose and high insulin”.
In the results figure 6 why no data about proliferating and apoptotic cells in the ICM and TE in case of high insulin condition, please clarify?
Indeed, we decided not to perform cell counting for the HI group for several reasons. First, we did not observe differences in gene expression related to cell cycle in HI group. Moreover, the HI blastocyst rate (Table 1) and total cell number (Suppl fig1) were not affected.
Quantification of apoptosis and proliferation on ICM and TE requires many embryos. Good development of the embryos to the blastocyst stage is required. Immunosurgery must be successful. The number of embryos for which ICM and TE are correctly sampled is small compared to the number of 1-cell embryos at the beginning of the culture. Depending on the period of the year, the number of embryos collected per rabbit is very variable. Despite the ovarian stimulation performed to increase the number of embryos collected per rabbit, many rabbits are needed to perform these experiments. To respect the 3Rs principle as much as possible, as developmental rate and gene expression did not suggest differences, we did not choose to quantify proliferation and apoptosis in HI ICM and HI TE.
The results should be accompanied by representative images for fluorescence microscopy, DAPI staining, and TUNEL apoptosis assay.
Representative merge images of TUNEL and EdU assays were previously presented in the Supplementary Figure 2 and 3. We have included DAPI, TUNEL or EdU and merge images. EdU assay in ICM is now presented in Supplementary Figure 2; EdU assay in TE is now presented in Supplementary Figure 3; TUNEL assay in ICM is now presented in Supplementary Figure 4; TUNEL assay in TE is now presented in Supplementary Figure 5.
The authors must explain how they selected these glucose and insulin doses for the current experiment.
Glucose and insulin concentrations were chosen based on previous studies in preimplantation embryos from different species exposed to glucose or insulin. Glucose concentration (15 mM) was chosen as an intermediate concentration from previous studies, notably Ramin et al., 2010 (doi:10.1210/en.2010-0187), Fraser et al., 2007 (doi:10.1093/humrep/dem318) and Leunda-Casi et al., 2001 (doi:10.1007/s001250100633). As for insulin, fewer studies have assessed insulin exposure during preimplantation development. We based our insulin concentration (1.7 µM) from the study of Laskowski, et al., 2016 (doi:10.1071/RD15315), which demonstrated that oocyte maturation in a high insulin environment in vitro impaired the developmental competence and gene expression patterns in the bovine.
Title should be very clear in terms of in vitro experiment and abbreviation should be avoided in the title.
Thank you for this suggestion. Title has been changed to: “In vitro exposure to glucose and insulin during the preimplantation period in the rabbit: Identification of the inner cell mass and the trophectoderm responses”
Please correct the manuscript for language and grammatical mistakes.
We hope that now the mansucript does not contain any mistakes.

Reviewer 2 Report
The manuscript from Via Y Rada et al. aims to observe differences in transcripts from the inner cell mass (ICM) or from the trophectoderm (TE) of rabbit embryos exposed to higher glucose concentrations, with or without insulin. The manuscript provides additional detail on the observation that increased glucose can negatively affect ICM or TE.
Although interesting, there are some points that can be improved in this manuscript.
The main concern is sampling. How can one guarantee that there is no contamination of cellular populations? Although clustering reveals that ICM samples and TE samples do not cluster together, it is prudent to acknowledge this in the discussion.
Still discussing sampling, considering the experiments that used cell counts after mechanical isolation of ICM and TE: How can one guarantee that no cells were lost in the mechanical isolation? Also consider again the issue of contamination of cellular populations. Dealing with such a small cell population, any loss or addition of cells could affect the final outcome of the statistical analysis. I suggest adding cell counts from whole embryos, using immunofluorescence for SOX2 and/or CDX2 combined with DAPI to determine ICM/TE cells and total cells. Especially considering that whole embryos were fixed for total cell count, immunostaining could have been performed. This would provide a more meaningful result than the proliferation assay. The EdU assay provides a snapshot of what is occurring at the blastocyst stage, but what about the effects throughout the experiment?
Another point is the analysis. The way I see the experiment, it is a 2 x 2 factorial (low/high glucose x with/without insulin). It seems that it was not considered like this, as the interaction of glucose and insulin was not considered in the statistical model. Analysis of data considering this design could infer if there were effects attributed exclusively to glucose concentration or to insulin.
Perhaps this type of analysis could facilitate writing of the results section, which is confusing as it goes back and forth between figures and tables. Perhaps a reorganization of sections based on experimental methods, not on experimental groups, could be easier to follow. Then, comparison of experimental groups could be described within these sections.
A specific point, but as important, are the TUNEL results. TUNEL assay images show arrows that are not pointing to cell nuclei, especially on the control group images. Also, why TUNEL is strongly labeling the whole cells in other images? Based on these images, TUNEL assay does not seem to be working properly. I would replace it with active or cleaved caspase detection. The latter can be performed with antibodies, allowing a double stain with SOX2 or CDX2 to determine apoptosis in TE or ICM cells.
The discussion also could be reorganized. Like the results section, there is a lot of back and forth. For example, it goes from high glucose in TE to high insulin then high glucose again. Another example, mTOR signaling in the TE is discussed twice. It would be ok if a different topic such as cell proliferation was clearly stated before the text shifted gears.
Specific comments
The introduction would benefit if a more assertive hypothesis was added. What authors truly expected to occur based on the presented literature? Not only mention that the cells may be affected.
Please describe in the methods section how many samples were submitted to sequencing. From the results it can be inferred that there were three samples per group.
Since the z-axis was used to obtain images, how slides were prepared? Was there any spacer?
Table 1 could suppress the concentrations of glucose and insulin as they are described in the methods section. The table seems a bit large as it is. Why the control group is in bold font and separated from the others?
The names of the groups could be added to the colored rectangles in figure 1 to reinforce the adopted nomenclature.
Figure 3 - Since it looks like a factorial design and, based on PCA graphs that show HG and HGI groups grouped closer to each other, it could be interesting to observe comparisons between low glucose (CTRL+HI) vs. high glucose (HG+HGI) conditions and low insulin (CTRL+HG) vs. high insulin (HI+HGI), provided there are no effects of the interaction between them.
Figures 4, 5 and 9 are not very useful as they are hard to read even if zooming in. They could be presented as supplemental figures instead. Or presented somehow as a shortened version of table 2, listing the number of enriched pathways present in each SUMER cluster.
Please describe NES (normalized enrichment scores) in the main text.
There are some passages in bold or underline.
Figure 6 - Why no cell counts for the HI group were performed? Although no differences were observed in gene expression related to cell cycle, those should be added for a clearer view of the of insulin effects.
Figure 6 - The y-axes are confusing. Since the Y-axis title states the percentage of positive cells, if I am perceiving it correctly, the values should show, for example in panel A: 20, 40 and 60%. Especially for the TUNEL panels, if these numbers are depicted correctly as 0.01%, the average number of cells per embryo would be lower than 0.1.
Figures 7 and 8 could be combined into one.
Page 18 - Split paragraph in the first line. It is too long and not on the same subject.
Author Response
Please see in the attachment

Round 2
Reviewer 1 Report
Even though the authors responded to the majority of the comments, a few small adjustments still need to be addressed.
1. I strongly suggest the authors to change the title as follows: Identification of the inner cell mass and the responses of the trophectoderm after in vitro administration of glucose and insulin to a rabbit embryos during the preimplantation period.
2. Although there is information about type 2 diabetes, which is characterized by insulin resistance, in the introduction, the author should only concentrate on type 1 diabetes or the consequences of hyperglycemia that are appropriate for the experimental design and the study's objectives.
3. Total cell number quantification in methods not clear how the cells counted or quantified. Please clarify.
Author Response
- Thank you for your title proposition. The title was rewritten in one sentence, avoiding the use of " : ". As suggested by the reviewer, we reversed the sentences to begin with “Identification of the inner cell mass and the trophectoderm…”. However, we have maintained the use of the term “exposure”, which seems more appropriate for in vitro experiments, as opposed to the term “administration”, which refers more to in vivo experiments.
- Several references to T1D hyperglycemia are present in the introduction, thanks to the reviewer #1. The aim of our work was to investigate the effects of deregulation of glycemia and insulinemia, these parameters are present in T1D, but especially in unmanaged T1D, which is quite rare in women of childbearing age in developed countries. These parameters are frequent in the early stages of prediabetic metabolic disorders, hence our more focused reference to T2D.
- Total cell number quantification in methods has been clarified.
Reviewer 2 Report
The manuscript from Via y Rada et al. is improved from the first version, but not so much. The description of the results is still confusing, although pie charts significantly improved the quality of pictures.
I believe authors should include in the manuscript that immunostainings were performed (include antibodies and catalog numbers) for whole embryos but results were inconsistent and thus not presented. If not included, this will be a question in most readers' minds.
Authors added hypotheses, however they are still not assertive as suggested. I still believe that the lack of an assertive hypothesis impacted the description of results and therefore, the hard-to-follow text.
The lack of analysis for cell counts in HI embryos was not clear for me. Based on your table 1, there were 530 embryos treated with high insulin. 64.5% of them turned into blastocysts, which results in 341 embryos. 11-16 embryos were used for RNAseq (341-16= 325 left). The numbers of embryos used for whole cell count, TUNEL and EdU are not clear, as just the number of experiments are reported (5-9 and 2-6, respectively). I think 325 embryos is still a good number of embryos to perform all experiments, including cell counts.
But what draws most attention in this response is that authors stated that: "The number of embryos for which ICM and TE are correctly sampled is small compared to the number of 1-cell embryos at the beginning of the culture". Which brings me back to the very main point of my first review, the issue with sampling using this technique. My suggestion was that authors acknowledge this issue in the discussion, especially when dealing with cell count data. And this is further confirmed by another comment from the authors: " In order to get a good separation with as little contamination as possible, we surely lost some cells, mainly TE. We can't guarantee that no cells were lost in the mechanical isolation".
Anyway, don't get me wrong; I just think it is important to acknowledge this flaw in the discussion section, again, especially when considering cell count data. It does not invalidate the experiment, especially the RNAseq which is supported by the PCA clustering.
Considering the statistical analysis, I believe that data could be presented differently, especially now knowing that there is no interaction between glucose and insulin, which would highlight the effects of each of the factors in the experiment. But, it is up to the authors.
Author Response
- Details on the immunostaining performed have been added in the method section. Thank you for this suggestion
- Number of embryos used for whole cell count has been added in the results section. Number of ICM and TE used for TUNEL and EdU experiments can be found in the legend of Fig 6. Proliferation and apoptosis quantifications have been performed after RNA-seq analyses. As we did not observe differences in cell cycle related gene expression and cell number, we did not prioritize the HI condition when we performed these quantifications. These experiments require a lot of animals and at this period, we faced difficulties in obtaining pregnant rabbits in our animal facility.
- The notion of a potential contamination has been added in the first paragraph of the results section.